# Prognostic Value of the Immunological Subtypes of Adolescent and Adult T-Cell Lymphoblastic Lymphoma; an Ultra-High-Risk Pro-T/CD2(−) Subtype

**DOI:** 10.3390/cancers13081911

**Published:** 2021-04-15

**Authors:** Beata Ostrowska, Grzegorz Rymkiewicz, Magdalena Chechlinska, Katarzyna Blachnio, Katarzyna Domanska-Czyz, Zbigniew Bystydzienski, Joanna Romejko-Jarosinska, Anita Borysiuk, Sebastian Rybski, Wojciech Michalski, Jan Walewski

**Affiliations:** 1Department of Lymphoid Malignancies, Maria Sklodowska-Curie National Research Institute of Oncology, 02-781 Warsaw, Poland; katarzyna.domanska-czyz@pib-nio.pl (K.D.-C.); joanna.romejko-jarosinska@pib-nio.pl (J.R.-J.); jan.walewski@pib-nio.pl (J.W.); 2Flow Cytometry Laboratory, Department of Pathology and Laboratory Diagnostics, Maria Sklodowska-Curie National Research Institute of Oncology, 02-781 Warsaw, Poland; katarzyna.blachnio@pib-nio.pl (K.B.); Zbigniew.Bystydzienski@pib-nio.pl (Z.B.); anita.borysiuk@pib-nio.pl (A.B.); 3Department of Cancer Biology, Maria Sklodowska-Curie National Research Institute of Oncology, 02-781 Warsaw, Poland; magdalena.chechlinska@pib-nio.pl; 4Department, of Mathematical Oncology, Maria Sklodowska-Curie National Research Institute of Oncology, 02-781 Warsaw, Poland; sebastian.rybski@pib-nio.pl (S.R.); wojciech.michalski@pib-nio.pl (W.M.)

**Keywords:** CD2, T-LBL, T-ALL/LBL, lymphoblastic lymphoma, flow cytometry

## Abstract

**Simple Summary:**

T-cell lymphoblastic lymphoma (T-LBL) is extremely rare and aggressive with no practical risk model defined. Considering the controversies over the prognostic value of T-LBL immunological subtypes, we re-evaluated 49 subsequent adult T-LBL patients treated according to the German Multicenter Study Group for Adult Acute Lymphoblastic Leukemia protocols, with 85.7% with complete remissions. To the best of our knowledge, this is the largest study of T-LBL diagnosed by flow-cytometry of the material obtained by fine-needle aspiration biopsy. We show that (1) CD2 status and age are powerful independent prognostic factors influencing overall survival and the risk of treatment failure; (2) the early/pro-T/CD2(−) subtype is associated with extremely poor outcomes; and (3) poor outcomes in ETP vs. non-ETP are strikingly consistent with the pro-T CD2(−) subtype. The lack of CD2 expression in T-LBL emerges as a new marker of an ultra-high-risk of treatment failure. We show here that ETP is a non-uniform entity, where the outcome depends on the CD2 status.

**Abstract:**

(1) Background: T-cell lymphoblastic lymphoma (T-LBL) is extremely rare and highly aggressive, with no practical risk model defined yet. The prognostic value of T-LBL immunological subtypes is still a matter of controversy. (2) Methods: We re-evaluated 49 subsequent adult T-LBL patients treated according to the German Multicenter Study Group for Adult Acute Lymphoblastic Leukemia (GMALL) protocols, 05/93 (*n* = 20) and T-LBL 1/2004 (*n* = 29), 85.7% of which achieved complete remission (CR). (3) Results: The 5/10-year overall survival (OS) and event-free survival (EFS) were 62%/59% and 48%/43%, respectively. In 96% of patients, flow cytometry analyses defining the WHO 2008 immunophenotypes were available. Cortical, early/pro-T/CD2(−), early/pre-T/CD2(+), and mature subtypes were identified in 59.5%, 19%, 15%, and 6.5% of patients, respectively. Overall, 20% of patients had the early T-cell precursor (ETP)-LBL immunophenotype, as proposed by the WHO 2017 classification. For the early/pro-T/CD2(−) subtype, the five-year OS and EFS were 13% and 13%, while for all the other, non-pro-T subtypes, they were 69% and 67%. By multivariate analysis, only CD2(−) status and age > 35 years emerged as strong, independent factors influencing OS and EFS, while the risk of CR failure was influenced by age only (>35 years). (4) Conclusions: ETP was non-significant for OS, unless an ultra-high-risk pro-T/CD2(−) subtype was concerned.

## 1. Introduction

T-cell lymphoblastic lymphoma (T-LBL) is an aggressive, very rare malignancy from precursor thymic T cells transformed at different stages of differentiation. According to the definition given by the WHO, T-LBL is classified together with acute lymphoblastic leukemia (T-ALL) as T-ALL/LBL [1,2]. The current standard of treatment for adult LBL patients is the ALL-like intensive therapy [3,4]. Despite an overall complete remission (CR) rate of 76–93%, one-third of patients relapse, and the 5-year overall survival (OS) rate for T-LBL is 51–69% in prospective studies [5,6,7,8,9] and 42–48% according to national epidemiology databases [10,11]. In ALL, the risk factors of treatment failure have been quite well recognized and allow, among others, to qualify patients for allogeneic stem cell transplantation (allo-SCT). Contrary to this, in LBL, no practical risk model has yet been defined. The main role in an ALL risk model is attributed to the level of minimal residual disease (MRD), but for LBL patients, MRD tracking is limited [7]. Most oncologists and hematologists regard immunological subtypes as important prognostic factors, with cortical CD1a(+) T-ALL/LBL having a considerably better prognosis than early-T or mature T-ALL/LBL [12]. Most published data distinguish immunological subtypes considering the status of CD1a and surface (s)CD3 only, but not the CD2 status, and as a result the pro-T/CD2(−) and pre-T/CD2(+) subtypes fall into a single, early subtype (CD1a−/sCD3−) [12,13,14]. In contrast to T-ALL, few data are available on the incidence and prognostic value of immunologic subtypes of adolescent/adult T-LBL [9,15]. The prognostic impact of immunological subtypes may be especially significant in T-LBL, where most often there is no possibility to follow the MRD. Recently, a subtype of T-ALL/LBL derived from thymic cells at the early T-cell precursor (ETP) differentiation stage has been recognized as having an extremely poor prognosis [2,13,16,17,18,19]. Considering the uncertain prognostic value of T-LBL immunological subtypes in the context of the 2008 and 2017 WHO classifications [1,2], we reevaluated 49 consecutive adolescent/adult patients. We analyzed their outcomes along with the expression of differentiation markers to examine the influence of immunological subtypes on prognosis. All T-LBL patients followed frontline regimens according to the German Multicenter Study Group for Adult ALL (GMALL) protocols [5,9]. Flow cytometry (FCM) immunophenotyping of the cellular suspension, obtained by fine needle aspiration biopsy (FNAB) or by ultrasound-guided or computed tomography (CT)-guided-FNAB from the involved lymph nodes or mediastinal tumors, was performed.

## 2. Materials and Methods

Forty-nine adult patients with T-LBL treated at the Maria Sklodowska-Curie National Research Institute of Oncology between 2000 and 2018 were enrolled in the analyses. All patients met the diagnostic criteria of T-LBL, based on the identification of a neoplastic proliferation of blasts with cytoplasmic CD3 (cCD3) expression and usually also a nuclear expression of terminal deoxynucleotidyl transferase (TdT). A cutoff of <25% bone marrow (BM) blasts was used to define LBL [1,2]. The diagnosis of T-LBL was made by an expert hematopathologist, routinely performing FNAB/FCM, assessing MRD, and undertaking histopathology (HP) and immunohistochemical (IHC) examinations. In 47 patients, the immunological subtype was defined according to the 2008 WHO classification as pro-T (cCD3+/sCD3−/CD2−/CD7+/ CD1a−/CD4−/CD8−/CD34+/−), pre-T (cCD3+/sCD3−/CD2+/ CD7+/CD1a−/CD4−/CD8−/CD34+/−), cortical T (cCD3+/sCD3−/CD2+/CD7+/CD1a+/ CD4+/CD8+/CD34−), or medullary T (cCD3+/sCD3+/CD2+/CD7+/CD1a−/CD34−/CD4+ or CD8+) [1]. The ETP subtype was defined according to the 2017 WHO classification and was distinguished by the following immunophenotype: CD1a(−)/CD8(−) and CD5(−) or CD5(+)^weaker^ with stem cell or myeloid marker expression [2,16]. The immunophenotype was determined by the FCM of cellular suspensions obtained before treatment from the involved lymph nodes (*n* = 21), mediastinal mass (*n* = 12), or nasopharyngeal/perimandibular infiltration (*n* = 2) by FNAB, and also from BM (*n* = 2), peripheral blood (PB) (*n* = 1), and pleural fluid (*n* = 7). Four to ten separate needle passes within a lymph node or tumor (three or four passes within mediastinal mass) provided adequate cellular material. FNABs were carried out with needles no. 21–23G and syringes rinsed in physiological saline (PBS) with K_2_ EDTA (ethylenediaminetetraacetic acid). A minimal panel of anti-human monoclonal antibodies (mAb) used for the FCM analysis to evaluate T-ALL/LBLs and ETP included antibodies against CD(1a/2/c3/s3/4/5/7/8/13/15/33/34/c79α), MPO/TdT, and HLADR (Appendix A). Additional mAbs used for extended FCM diagnostics are included in the Appendix A and listed in Appendix A. Because of immunophenotype diversity related to the ontogenetic maturity of T-LBL, within one half-hour before the full panel was tested, we assessed a basic lymphoma panel (an initial test) in order to set the ultimate individual panel for each patient. FNAB samples were prepared by a standard technique of 3- or 4-color immunophenotyping as previously described [20]. For T-LBL diagnosis, cells obtained by FNAB of the lymph nodes/tumors were immunophenotyped as previously described [20]. For CD2 expression, PE-conjugated CD2 antibody clone S5.2, Becton Dickinson Biosciences, BD, was used; all mAbs are specified in Appendix A. Antigen (Ag) expression was quantified by FACScan (until 2000y), FACSCalibur (until 2007y), and FACSCanto II (to this day) cytometers (BD, San Jose, CA, USA), and categorized into three groups according to the percentages of positive cells: “(−)”, not expressed, if detected in < 20% of neoplastic cells; “(+/−)”, if expressed in ≥ 20% and < 100% of neoplastic cells; “(+)”, if expressed in 100% of lymphoma cells. Expression was quantified as (+)^weaker^ or (+)^higher^ than that on control T lymphocytes, as previously described [20,21]. Cells obtained by FNAB were also stained with a rapid hematoxylin and eosin stain, and smears performed from BM/PB were stained with the May–Grünwald–Giemsa for morphological evaluation. In 47 patients diagnosed by FNAB/FCM, 43 patients were diagnosed also by HP/IHC of trephine bone marrow and surgical biopsies. On top of that, two patients had HP only with a limited IHC panel, without FCM. Between 2000 and 2006, immunohistochemical diagnosis of T-LBL was based on the expression of a limited number of antigens, i.e., CD3, CD34, TdT, CD20, Ki-67, and MPO. In 2007, the IHC panel was supplemented with CD1a, CD2, CD4, CD5, CD7, and CD8, necessary to assess the T-LBL immunological subtypes. In 10 patients, CD2 was assessed by both immunohistochemical staining and FCM (four CD2-negative and six CD2-positive). An IHC reaction for CD2 was considered positive if any of the T-LBL cells showed staining, against control normal T lymphocytes which are always CD2(+)^higher^. The IHC results were consistent with FCM in all cases. The diagnostic material taken by FNAB, if sufficient, was also sent for cytogenetics, as previously described [20]. Patients were treated according to GMALL 05/93 (years 2000–2004) and GMALL T-LBL 1/2004 (years 2005–2018) protocols (Appendix A) [5,9]. CR was defined as a resolution of extramedullary disease, normal BM status (≤5% blasts), neutrophil count ≥1.0 G/L, and platelet count ≥ 100 G/L. The mediastinal mass response was assessed by CT imaging. It was considered residual if there was a decrease in dimension of over 75% that was stable in two subsequent CT examinations. Since September 2010, positron emission tomography/computed tomography has been routinely used to confirm CR after second induction (seventh week of treatment).

### Statistical Analysis

OS was calculated from the date of treatment initiation to death or to the date of the last follow-up in surviving patients. Event-free survival (EFS) was calculated from the beginning of treatment until an event (relapse, treatment failure, death during induction, or death during complete remission). Patients with stem cell transplantation (SCT) were censored at the time of transplantation. Survival curves were calculated by the Kaplan–Meier method and compared with the log-rank test. Differences of frequency were analyzed with the Chi-squared or Fisher exact test. Univariate and multivariate analyses were performed to identify prognostic factors (age, white blood cell (WBC) count, hemoglobin, platelet count, central nervous system (CNS) involvement at diagnosis, BM involvement at diagnosis, GMALL 05/93 vs. GMALL 2004 treatment, WHO classification (early/pro-T, early/pre-T, cortical, mature) and ETP vs. non-ETP, expression of CD(1a/2/s3/4/8/5/7) vs. non-expression, and the number of pan-T Ags expressed (0–3 vs. 4–7). The Cox Proportional Hazard Model (PHM) was used for factor analysis of the main endpoints, and the logistic regression model was applied for secondary endpoint analysis. A significance level of *p* < 0.05 was set for all statistical tests. The IBM SPSS 23 package was used for statistical analysis.

## 3. Results

Between 2000 and 2018, 49 adult patients with T-LBL were treated according to the GMALL 05/93 (*n* = 20) and GMALL T-LBL 1/2004 (*n* = 29) protocols. The median age of patients at diagnosis was 28 years (range 16–56), with 35 (71.4%) younger than 35 years, and the majority of patients were males (75.5%). The most typical location of the disease was bulky mediastinal mass (92%), with simultaneous pleural effusion in almost half of the patients (47%) and pericardial effusion in one third (35%). The mean WBC count was 9.9 G/L (range, 2.5–21.2), HGB 13.7 g/dl (range, 9.1–17.7), and PLT 313 G/L (range, 151–788). Based on the HP assessment of trephine bone marrow biopsy specimens, bone marrow involvement was found in 22% (*n* = 11) of patients. Classical trephine bone marrow biopsy and FCM evaluations were performed simultaneously in 15 patients. In nearly half of them (7/15 (46.6%)), low BM involvement was evidenced by FCM, with no evidence of involvement in classical trephine bone marrow biopsy. Primary CNS involvement concerned only three patients (6%). The baseline characteristics of the T-LBL patients are detailed in Appendix A. Overall, 42 (85.7%) of the patients achieved CR. The median follow-up was 155.6 months. The 5/10-year OS and EFS were 62%/59% and 48/43%, respectively. Forty-seven patients (96%) had a comprehensive FCM analysis to define subtypes according to the 2008 WHO criteria for T-ALL/LBL, based on the status of the pan-T Ag panel: CD1a, CD2, sCD3, CD4, CD5, CD7, and CD8. The most common T-LBL was of the cortical subtype (59.5%), followed by the early subtype (34%), subdivided by CD2 status to the early/pro-T/CD2(−) (19%), and the early/pre-T/CD2(+) (15%), while the least frequent was the mature subtype (6.5%). In part of cases, we managed to set the karyotype. A complex karyotype was identified in 7/8 (88%) CD2(−) cases and 10/18 (56%) CD2(+).

The 5/10-year OS and EFS were strongly related to the 2008 WHO subtypes. For the cortical subtype, 5/10-year OS and EFS were 75%/69% and 75%/68%, while for the non-cortical subtypes they were 38%/38% and 31%/31%, respectively (*p* = 0.023/0.002) (Figure 1).

For the early/pro-T subtype, the 5-year OS and EFS were 13% and 13%, while for all the other non-pro-T they were 69% and 67%, respectively (*p* = 0.001/< 0.001) (Figure 2).

The five-year OS for patients with CD2, CD1a, and more than three of the pan-T Ag panel expressions was 73%, 78%, and 76%, compared to 26%, 34%, and 28% for patients without CD2 and CD1a and no more than three of the pan-T Ag expressions, respectively (*p* = 0.001/ = 0.002/ = 0.001) (details in Table 1, Figure 3 and Figure 4).

By univariate analysis, the following variables significantly influenced survival: age (≤35 vs. >35); the number of pan-T Ags (4–7 vs. 0–3 present); and CD1a, CD2, sCD3, and CD8 present vs. absent. By multivariate analysis, only age >35 years (hazard ratio (HR), 5.39; 95% confidence interval (CI), 2.10–13.8; *p* < 0.001) and the lack of CD2 expression (HR, 5.10; 95% CI, 1.93–13.49; *p <* 0.001) were powerful independent prognostic factors, influencing OS. Similarly, age > 35 years and CD2(−) status were the only significant factors for EFS (*p* = 0.001 and *p* < 0.001, respectively), details in Table 2.

There was a significant correlation between CD2 expression and the expression of CD1a (*p* < 0.001), sCD3 (*p* = 0.002), CD4 (*p* = 0.002), CD8 (*p* < 0.001), and of more than three pan-T Ags (*p* < 0.001), the cortical subtype (*p* < 0.001), and subtypes other than the pro-T (*p* < 0.001). Overall, 27.6% (*n* = 13) of patients were CD2(−), including 69% (*n* = 9) of patients with the early/pro-T subtype (with 2–3 of pan-T Ags only), two patients with the medullary subtype, and two patients with an unusual CD2(−) cortical subtype. Nearly half (*n* = 6) of the CD2(−) patients met the criteria of the ETP subtype. The median age was 30 years (range, 18–57), with eight (61%) younger than 35 years; male sex was dominant (92%). Only 69% (*n* = 9) of the CD2(−) patients achieved CR, compared with 91% (*n* = 31) in the CD2(+) patients, and 77% (*n* = 10) progressed, compared with only 20.5% (*n* = 7) in the CD2(+) patients. Of the 13 CD2(−) patients, five (38.5%) received SCT, among them two in the first-line treatment (CR1) and three in the second remission (CR2). All four CD2(−) patients who are alive (30.7%) have undergone SCT: one in CCR1, now three years after allo-SCT, and the other three in CCR2 (one of those relapsed 11 years after CR1, was transplanted in CR2, and now it is his fifth year without evidence of disease; the other two are without evidence of disease 11 and 3 years after allo-SCT). Baseline characteristics and outcomes for CD2(−) vs. CD2(+) are detailed in Table 3 (47 patients diagnosed by FNAB/FCM) and Appendix A (49 patients, including two with CD2 expression detected by IHC).

Overall, 20% (*n* = 9) of patients had ETP-ALL/LBL immunophenotype, with CD5(−) (44.4%) or CD5(+)^weaker^ (55.6%) expression, co-expressed with CD34(100%), HLA-DR(37.5%), CD13(40%), CD33(80%), and CD15(25%). According to the 2008 WHO subtyping, three patients (33%) with ETP were categorized as ETP/pre-T/CD2(+) and six patients (67%) as ETP/pro-T/CD2(−). There was no statistically significant difference in OS (*p* = 0.135) among patients with the ETP vs. non-ETP subtype, with a five-year OS of 33% and 66%, respectively, and a five-year EFS of 26% and 59%, respectively (*p* = 0.095) (Figure 5).

Among nine ETP patients, three (33%) are alive to date. They include two out of the three patients with ETP/pre-T/CD2(+) and only one of the six patients with the ETP/pro-T/CD2(−) phenotype. Overall, 44.4% (*n* = 4) of the ETP patients underwent SCT, two in the first line treatment (CR1), one in the second remission (CR2), and one in partial remission (PR/(MRD+) after CR1 failure. Two of the three patients who are alive received SCT.

## 4. Discussion

T-LBL is a very rare disease with only few publications on its clinical characteristics and treatment outcomes. In contrast to leukemia (T-ALL), for T-LBL no risk model stratification has yet been defined. No clinical features or laboratory markers have been widely accepted as prognostic factors, except patients’ age, usually defined as under or over 40 years; its prognostic significance is attributed to a lower tolerance of intensive chemotherapy in older patients. A prognostic model based on the rearrangements of four-genes, NOTCH1/FBXW7/RAS/PTEN, is important, but still difficult to apply in daily practice [4,8]. The role of immunophenotyping to determine blast maturity has also been recognized by the WHO 2008 classification, which distinguished four T-ALL/LBL ontogenetic subtypes. The WHO 2017 revision included a new provisional entity, an ETP subtype associated with very early ontogenesis [1,2]. Our study describes the immunophenotype distribution of T-LBL, classified according to the WHO 2008/2017 definitions, and the prognostic significance of an immunophenotype in the context of OS and EFS. We also analyzed the clinical features and outcomes in several T-LBL cases subjected to SCT. The clinical features of our patients did not differ from those described in most publications [5,6,7]. Primary CNS involvement was found in only three patients (6%) and had no prognostic significance for treatment outcomes. Bone marrow involvement, based on HP assessment of trephine bone marrow biopsy specimens, was found in 22% of patients. Evaluation of bone marrow by FCM identified minimal BM involvement undetectable by trephine biopsy in nearly half of the cases. This indicates that flow cytometric examination is a more sensitive method for MRD detection in T-LBL. However, the role of MRD for T-LBL is uncertain [7]. In this study, all patients with minimal disseminated disease (MDD) obtained BM eradication, and BM involvement, as shown by uni/multivariate analyses, proved not to influence the outcomes. Only the age of >35 years was a powerful independent prognostic factor, influencing OS, EFS, and the risk of CR failure. Recently, we have demonstrated that in different types of aggressive lymphomas, FNAB/FCM presents a better diagnostic accuracy and effectiveness than the routine HP/IHC examination [20,21]. Therefore, at our institution, in all cases clinically suspected of T-ALL/LBL or with HP/IHC-confirmed T-ALL/LBL, we always aim to perform FNAB/FCM. In contrast to T-ALL, few data are available on the incidence or prognostic value of immunologic subtypes of adult T-LBL. Here, we report a series of consecutive 49 adult patients with T-LBL, and in 47 of these cases a comprehensive FCM analysis was performed. To the best of our knowledge, this is one of the largest published cohorts of T-LBL cases, including ETP-LBL cases (20%) diagnosed by the FNAB/FCM.

The cortical subtype was found to be the most common, followed by the pro-T/CD2(−), pre-T/CD2(+), and medullary subtypes. In the largest group of 105 prospectively collected adult patients with T-LBL (GMALL) [9], the incidence of cortical and early or mature phenotypes was similar to those in our study. Contrary to this, according to a recent publication from China, among 64 T-LBL patients with FCM results, only 8% was of the cortical subtype, and the most frequent were the early and mature/medullary subtypes (70% and 22%, respectively) [15]. In our previous study, we showed no differences in the subtype frequency between LBL vs. ALL presentation [22,23]. In accordance with our data, a correlation between the stages of T-cell differentiation and survival, mainly in T-ALL, has been shown. Based on the results of GMALL 05/93 study, early and mature T-ALL were recognized as high-risk subtypes, and allo-SCT was recommended in CR1 [14,24]. In contrast to GMALL, in MD Anderson Cancer Center (MDACC) protocols the routine use of allo-SCT in CR1 was not recommended based on immunological subtypes regarded as high-risk by GMALL. Jain et al. [13] found no difference in CR rates, OS, and EFS, between the early, cortical, and mature subtypes. However, their subtyping was based mainly on CD1a and sCD3 status, and the distinction between pro-T/CD2(−) and pre-T/CD2(+) has not been performed in most cases, so the proportion between the early/pro-T and early/pre-T subtypes was biased. In addition, the cortical subtype was recognized in only 24.5% of patients. Interestingly, the median OS of early/pre-T (CD2+) subtype patients was not reached. As recently presented by the Spanish PETHEMA group, the most favorable outcome in a T-ALL cohort was in the pre-T and cortical subtypes, with a four-year OS of 61% and 54%, respectively, while it was only 30% and 39% in the ETP and mature subtypes, respectively [18]. Unfortunately, data on the outcomes of T-LBL patients are limited. In the above-mentioned studies, the GMALL LBL cohort presented a five-year OS slightly superior for the cortical (78%) than for the early/mature subtype (58%), while the China LBL cohort showed no difference in progression-free survival (PFS) between the immunophenotype subgroups [9,15]. In our previous studies, the most favorable outcomes were related to the cortical and early/pre-T/CD2(+) T-ALL/LBL subtypes, with a five-year OS of 69% and 48%, respectively [22,23]. Considering the prognostic significance of CD2 expression, it is apparent that pro-T/CD2(−) and pre-T/CD2(+) cases should not be classified together as the early subtype. In our series of 47 T-LBL patients with comprehensive immunophenotyping, OS and EFS depended to a large extent on the 2008 WHO subtype. As expected, for the cortical (CD1a+) subtype, the 5/10-year OS and EFS were significantly better than those of the non-cortical subtypes. However, extremely poor outcomes were shown for the early pro-T/CD2(−) subtype as compared with all the other, non-pro-T subtypes. In the Cancer and Leukemia Group B (CALGB) series of patients treated according to the 8364 protocol, a number of T cell markers expressed in T-ALL cases was shown to be prognostically significant. In particular, patients expressing six or seven T cell antigens had a longer PFS and OS than patients expressing three or fewer antigens. In addition, patients with CD1a, CD2, CD4, and CD5 expression had significantly improved survival rates [25]. In our series, a five-year OS for patients with CD2, CD1a, and more than three pan-T Ags expressed was significantly higher than in patients with the lack of CD2 and CD1a and no more than three pan-T Ag expression. By multivariate analysis that incorporated the whole panel of antigens and numerous clinical data, negative CD2 status and age >35 years were the only and powerful independent prognostic factors influencing OS and EFS. To the best of our knowledge, the expression of CD2 in patients with T-ALL/LBL has not been clearly demonstrated as a prognostic factor. In the UKALLXII/ECOG 2993 study, an association between CD2 positivity and simple karyotypes was found in ALL/LBL. Patients expressing CD2 rarely had a complex karyotype (only 2%), while 27% of the CD2(−) patients did have a complex karyotype. At the same time, a complex karyotype vs. a simple/normal karyotype was significantly associated with a lower five-year OS [12]. Our data are in line with the UKALLXII/ECOG 2993 study and suggest that a complex karyotype is more frequent in CD2-negative cases. However, the number of our cases with a karyotype tested was too small to prove statistical significance. Other studies have shown that CD2 negativity correlated with the immature T-cell receptor and TCRγδ T-ALL lineages, which are associated with the occurrence of the CALM-AF10 fusion gene and the TLX3/HOX11L2 and MLL(KMT2A) gene rearrangements, considered to be prognostically unfavorable [26,27,28].

CD2 is a cell adhesion molecule found on the surface of T cells and natural killer (NK) cells. In humans, it interacts with other adhesion molecules, such as lymphocyte function-associated antigen-3 (LFA-3/CD58), which are expressed on the surface of other cells. In addition to its adhesive properties, CD2 also acts as a co-stimulatory molecule for T and NK cells [29,30]. One may speculate that the lack of CD2 expression on T-ALL/LBL cells minimizes interactions between neoplastic and immune cells, enabling relapse even from the MRD level. In our series, 27.7% of T-LBL was CD2(−). Although there was no major difference in the baseline characteristics between the CD2(−) and CD2(+) T-LBL patients, the CD2(−) patients showed a higher percentage of induction failure than the CD2(+) patients, and only 69% achieved CR, compared with 91% in the CD2(+) patients. Progression occurred in up to 77% of the CD2(−) patients, compared to only 20.5% for CD2(+). Only four (30.7%) of the CD2(−) patients are alive; all received allo-SCT. The majority (69%) of CD2(−) patients had the pro-T subtype, but the loss of CD2 expression is also rare in the other subtypes. In two cases, the lack of CD2 was identified in the cortical phenotype associated with an overall good prognosis, but both patients experienced recurrence of the disease in the CNS. One had a confirmed lymphoma infiltration in the spinal canal, the other had a minimal CSF involvement by FCM. Both patients have remained in CR2 after allo-SCT (for 11 and 3 years until now). Six out of thirteen (46%) of our CD2(−) patients met the criteria of the ETP subgroup.

Many cases previously classified as pro-T or pre-T would now meet the criteria for ETP. Among immunological subtypes, ETP is now the most widely discussed, but there is still controversy over the prognostic significance of the ETP T-ALL/LBL subgroup. In the MDACC series of T-ALL/LBL patients, 17% were recognized as ETP, and of the ETP patients for whom sufficient data were available, approximately half were categorized as early/pro-T and half as early/pre-T subtypes; only four patients with ETP had LBL presentation. ETP patients had significantly worse OS than patients with non-ETP ALL, with no difference in EFS [13]. Recently, in the Group for Research on Adult Acute Lymphoblastic Leukemia (GRAALL) 2003 and 2005 studies, 22.1% of patients with the ETP phenotype were treated with the pediatric-type regimen. Based on this intensive strategy, the overall prognosis of ETP-ALL was similar to that of the rest of the T-ALL cohort, with the impressive five-year OS of 59.6% and 66.5% for ETP and non-ETP, respectively [17]. An important difference was observed in the rate of SCT for patients in the first CR between the MDACC and GRAALL studies, with only 17.6% patients transplanted in MDACC and 48.9% in GRAALL. It was concluded by GRAALL that the ETP subgroup is a specific predictor of benefiting from allo-SCT [17]. On the other hand, the above-mentioned retrospective analysis by PETHEMA included 34 patients with ETP-ALL (20% of T-ALL cases) and revealed that treatment intensification with allo-SCT did not improve survival (with a five-year OS of 36%) [18]. Data from the Chinese study referred to above suggest that autologous SCT might overcome the unfavorable effect of the ETP-LBL subtype on prognosis [15]. In our series, 20% of the T-LBL patients were identified with ETP, a similar frequency to that found in the MDACC, PETHEMA, and GRAALL studies, but other than in the MDACC cohort, a higher percentage of our patients were categorized as pro-T-CD2(−)/ETP (66.6%) than as pre-T-CD2(+)/ETP (33.4%) [13]. In contrast to the CD2(−) patients as a whole subgroup, the ETP patients had a low risk of induction failure, with 88.8% of CR, while progression was twice as common in ETP (66.6%) compared to non-ETP (30.5%) patients. The ETP phenotype was not shown to influence OS and EFS, unlike the CD2(−) pro-T subtype. Among nine ETP patients, three are alive. They include two out of the three patients with ETP/pre-T/CD2(+) and only one of the six patients with the ETP/pro-T/CD2(−) phenotype. One could speculate that patients with ETP/pre-T/CD2(+) may have a better prognosis. In our small group of patients who had received allo-SCT, 50% (2/4) of the ETP patients and 80% (4/5) of the CD2(−) patients benefited from this procedure, but the optimal management remains unclear. It is likely that more targeted treatments are needed to improve the remission rate for the early T-ALL/LBL classified as ETP, and in our opinion also defined as CD2(−).

## 5. Conclusions

Our data show that (1) CD2 status, along with an age of less or over 35 years, is a powerful independent prognostic factor influencing OS and the risk of treatment failure; (2) the early/pro-T/CD2(−) subtype is associated with extremely poor outcomes, while all the other, non-pro-T subtypes show significantly better outcomes; (3) OS and EFS strongly relate to the 2008 WHO subtypes; (4) the cortical subtype is associated with a significantly better five-year OS and EFS than the non-cortical subtype; (5) poor outcomes in ETP vs. non-ETP are strikingly consistent with the pro-T/CD2(−) subtype.

To conclude, the lack of CD2 expression in T-LBL emerges as a new marker of an ultra-high-risk of treatment failure. This context casts doubts on the concept of ETP as a category of high-risk disease in all cases because, as we show here, ETP is a non-uniform entity, where the outcome depends on the CD2 status.

## Figures and Tables

**Figure 1 cancers-13-01911-f001:**
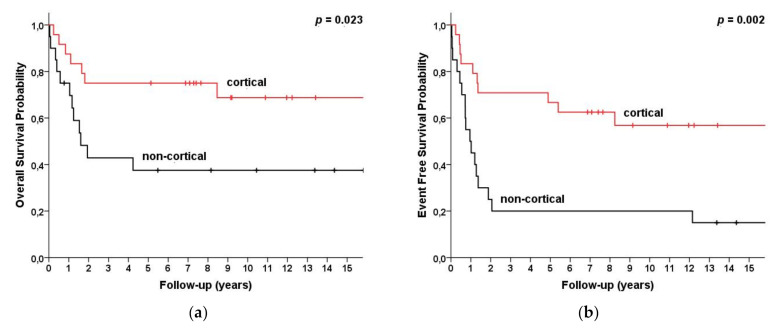
Survival for T-cell lymphoblastic lymphoma (T-LBL) (*n* = 47) patients categorized by immunological subtypes (WHO 2008) as cortical (*n* = 28) vs. non-cortical (*n* = 19) subtype. (**a**) Overall survival and (**b**) event-free survival.

**Figure 2 cancers-13-01911-f002:**
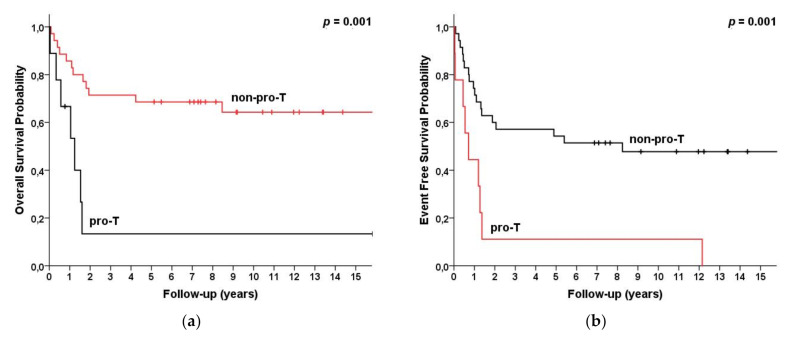
Survival for T-LBL (*n* = 47) patients categorized by immunological subtypes (WHO 2008) as early/pro-T (*n* = 9) vs. non-pro-T (*n* = 38) subtype. (**a**) Overall survival and (**b**) event-free survival.

**Figure 3 cancers-13-01911-f003:**
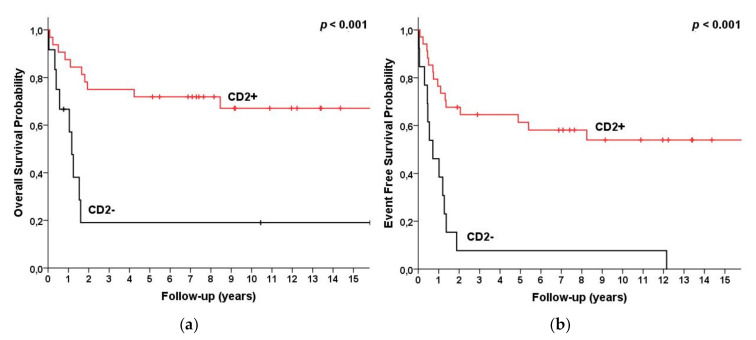
Survival for T-LBL (*n* = 47) patients categorized by CD2 status as CD2 present (*n* = 34) vs. CD2 absent (*n* = 13). (**a**) Overall survival and (**b**) event free survival.

**Figure 4 cancers-13-01911-f004:**
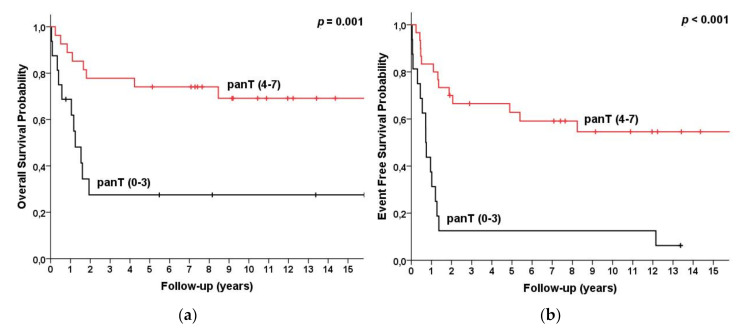
Survival for T-LBL (*n* = 46) patients categorized by number of pan-T Ags as pan-T 0–3 Ags present (*n* = 16) vs. 4–7 Ags present (*n* = 30). (**a**) Overall survival and (**b**) event-free survival.

**Figure 5 cancers-13-01911-f005:**
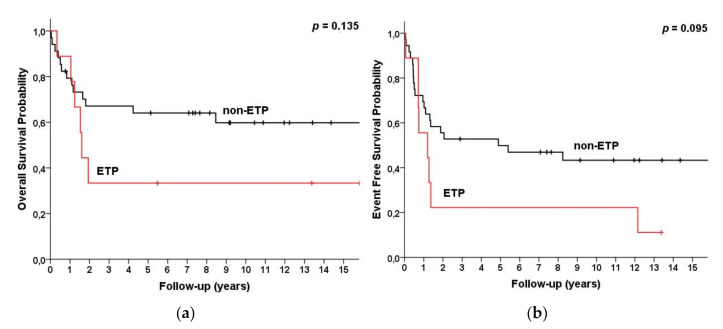
Survival for T-LBL (*n* = 45) patients categorized by subgroup defined by WHO 2017 as ETP (*n* = 9) vs. non-ETP (*n* = 36). (**a**) Overall survival and (**b**) event-free survival.

**Table 1 cancers-13-01911-t001:** Overall survival (OS) and event-free survival (EFS) according to age (*n* = 49), immunological subtypes and number of pan-T antigens (*n* = 47).

Variables	*n*	OS		*p*	EFS		*p*
		5 Year (%) (95% CI)	10 Year (%) (95% CI)		5 Year (%) (95% CI)	10 Year (%) (95% CI)	
	49	62 (48; 76)	59 (44; 74)	-	48 (33; 63)	43 (28; 58)	-
Age ≤ 35 Age > 35	35 14	76 (62, 91) 28 (5, 52)	76 (62, 91) 14 (0, 37)	<0.001	73 (58, 89) 26 (2, 50)	73 (58, 89) 13 (0, 34)	<0.001
Cortical Non-cortical	28 19	75 (58, 92) 38 (16, 59)	69 (49, 89) 38 (16, 59)	0.023	75 (58, 92) 31 (8, 54)	68 (48, 88) 31 (8, 54)	0.002
Pro-T Non-proT	9 35	13 (0, 38) 69 (53, 84)	13 (0, 25) 64 (47, 81)	0.001	13 (0, 38) 67 (51, 83)	13 (0, 38) 63 (45, 80)	<0.001
ETP Non-ETP	9 38	33 (3, 64) 66 (50, 82)	33 (3, 64) 61 (45, 78)	0.135	26 (0, 57) 63 (47, 80)	26 (0, 57) 59 (41, 76)	0.095
CD1a(+) CD1a(−)	28 19	78 (62, 94) 34 (12, 56)	72 (54, 90) 34 (12, 56)	0.002	76 (59, 93) 28 (6, 51)	69 (49, 89) 28 (6, 51)	0.001
CD2(+) CD2(−)	34 13	73 (58, 88) 26 (1, 51)	68 (52, 85) 26 (1, 51)	0.001	71 (55, 87) 19 (0, 43)	66 (48, 84) 19 (0, 43)	<0.001
0–3 panT Ags expressed	16	28 (5, 50)	28 (5, 50)	0.001	18 (0, 40)	18 (0, 46)	<0.001
4–7 panT Ags expressed	30	76 (61, 92)	71(54, 89)	0.001	74 (57, 72)	68 (49, 87)	<0.001

Abbreviations: ETP—early T-cell precursor; pan-T antigens—CD1a, CD2, sCD3, CD4, CD5, CD7, CD8.

**Table 2 cancers-13-01911-t002:** Results of multivariate analysis for OS, EFS, and CR endpoints.

CPHM	Variables	HR	95%CI	*p*
OS	Age > 35 CD2 negative	5.39 5.10	2.10–13.8 1.93–13.49	<0.001 0.001
EFS	Age > 35 CD2 negative	3.77 4.52	1.72–8.27 2.03–10.06	0.001 <0.001
**LRM**	**Variables**	**OR**	**95%CI**	***p***
CR	Age > 35 CD2 negative	0.039 0.193	0.004–0.413 0.025–1.50	0.007 0.116

Abbreviations: CPHM—Cox Proportional Hazard Model, LRM—logistic regression model, HR—hazard ratio, OR—odds ratio, CI—confidence interval, OS—overall survival, EFS—event-free survival, CR—complete remission.

**Table 3 cancers-13-01911-t003:** Baseline characteristics and outcomes for CD2(−) patients compared to CD2(+) patients diagnosed by fine needle aspiration biopsy (FNAB)/flow cytometry (FCM).

*n* (%)	Total *n* = 47 (100%)	CD2(+) *n* = 34 (100%)	CD2(−) *n* = 13 (100%)
Gender, male	36 (76.5%)	24 (70.5%)	12 (92%)
Age, median (range) <35 year	28 (18, 58) 33 (70%)	26 (18, 58) 25 (73.5%)	30 (18, 57) 8 (61%)
Treatment			
GMALL 05/93 GMALL 01/2004	18 (38%) 29 (62%)	13 (38%) 21 (62%)	5 (38.5%) 8 (61.5%)
Bone marrow involvement (<25%)	11 (23%)	7 (20.5%)	4 (30.5%)
CNS involvement	3 (6%)	2 (6%)	1 (7.5%)
	median (range)	median (range)	median (range)
WBC (G/L) HGB (g/dl) PLT (G/L)	9.2 (2.5–21.2) 13.8 (9.1–17.7) 317 (151–788)	8.6 (2.5–21.2) 13.4 (9.1–16.9) 296 (151–754)	10 (3–18) 14 (9.3–17.7) 349 (198–788)
Immunophenotype			
Pro-T Pre-T Cortical mature	9 (19%) 7 (15%) 28 (59.5%) 3 (6.5)	07 (20.5%) 26 (76.5%) 1 (3%)	9 (69%) 0 2 (15.5%) 2 (15.5%)
ETP *	9 (20%)	3 (9%)	6 (46%)
CR	40 (85%)	31 (91%)	9 (69%)
progression	17 (36%)	7 (20.5%)	10 (77%)
alive	28 (59.5%)	24 (70.5%)	4 (30.5%)

Abbreviations: GMALL—German Multicenter Study Group for Adult ALL; CNS—central nervous system; ETP—early-T precursor; WBC—white blood count; HGB—hemoglobin; PLT—platelets; CR—complete remission. * *n* = 45 patients.

## Data Availability

The data presented in this study are available in this article (and Appendix A).

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
