# Peer review of "Prognostic Value of the Immunological Subtypes of Adolescent and Adult T-Cell Lymphoblastic Lymphoma; an Ultra-High-Risk Pro-T/CD2(−) Subtype"

_cancers, 2021, doi:10.3390/cancers13081911_

Round 1

Reviewer 1 Report

Through a multivariant analysis, the authors demonstrated CD2-status is an independent prognostic factor for T-LBL patients. Also, they proposed ETP is a heterogeneous entity, and the poor outcomes in ETP are attributable to the pro-T CD2(–) subtype. It is a well-designed study and a well-written manuscript.

Major points:

  1. Did CD2- T-LBL patients in this cohort carry genetic abnormalities that have been proven to have a dismal prognosis? For example, a complex karyotype or particular gene mutations?
  2. Did the different treatment protocols, GMALL 05/93 and T-LBL 1/2004, contribute to the poor prognosis of CD2- T-LBL patients?

Minor points:

  1. Page 56, please rephrase, “In contrast to leukemia (ALL), where the risk factors of treatment failure have been quite well recognized, and allow, among others, to qualify patients for allogeneic stem cell transplantation (allo-SCT), for LBL no practical risk model has yet been defined.”
  2. Page 73, “we reevaluated 47 consecutive adult patients”. Should it be 49 consecutive patients?
  3. Page 90, “(cCD3+/sCD3/CD2−/CD7+/ CD1a−/CD4–/CD8–/CD34+/−)” should be “(cCD3+/sCD3-/CD2−/CD7+/ CD1a−/CD4–/CD8–/CD34+/−)”

Reviewer 2 Report

The manuscript ” Prognostic value of the immunological subtypes of adult T-cell lymphoblastic lymphoma, an ultra-high-risk pro-T/CD2(–) subtype” by Ostrowska et al. is a single-institution study on the impact of immunophenotypic subtypes (specifically, CD2 expression by flow cytometry) on prognosis and outcome in adolescents and young adult (AYA) patients with T lymphoblastic lymphoma (T-LBL).  The topic has practical utility and the authors present novel findings in this well-written study. 

The manuscript may benefit from addressing the following questions and suggestions, prior to further consideration:

  1. Title: The NCI defines adolescent and young adult patients (AYA) as 15-39 years old.  Since the patient cohort studied here had an age range of 16 to 56, and the majority (71.4%) of patients were younger than 35 years, it would be more accurate to describe it in the title and throughout the manuscript as “AYA” instead of “adult” T-LBL. 

  1. Material and Methods: The readers would be interested in knowing more details about the methods used in the study.  Can the authors confirm that 3- or 4-color flow cytometry was performed for all immunophenotyping?  If so, please provide a more detailed antibody combination (i.e. individual tube composition) and on how many patients that particular antibody panel was performed.  Since the patient cohort was accrued over almost two decades, please list the type of flow cytometers used over that interval, as well, and whether a retrospective review of the immunophenotypic data was performed in a blinded fashion by a single (or multiple?) Hematopathologist(s).  Also, please include the antigen expression criteria (how is positive expression defined by flow cytometry and immunohistochemistry) and indicated on how many cases CD2 expression was assessed both by FC and IHC.  For the 10 cases (7 body fluids, 2 bone marrows, 1 peripheral blood) that presumably had no concurrent immunophenotyping performed on a tissue biopsy, please confirm and state whether that was the fact. 

  2. Results: The authors indicated that 7/15 (46.6%) of patients had low bone marrow involvement by flow cytometry, with no evidence of disease in the core biopsy.  Please confirm that all patients had a staging bone marrow biopsy performed, including a manual differential count done on the aspirate smear to establish the blast percentage.  While there is typically high concordance between blasts percentages estimated by flow cytometry and manual differential count, only the latter is recognized as the “gold standard” for diagnostic purposes. 

  1. Discussion: Please see comment above related to bone marrow assessment.  One of the conclusions states that evaluation of bone marrow by flow cytometry identified minimal BM involvement undetectable in the core biopsy in nearly half of the cases.  It is well known that flow cytometric immunophenotyping is more sensitive than morphology by several orders of magnitude, in detecting minimal residual disease in T-ALL / LBL. 

Reviewer 3 Report

Dear All,

Thank you for asking me to review this interesting paper.  The finding of robust prognostic markers in context of newly diagnosed T-LBL is of interest.  While not mentioned in great detail in the introduction, some prognostic markers (genetic particularly) have indeed been suggested, these are mentioned in the discussion.   As also mentioned by the authors, those genetic markers may not be easy to employ for all laboratories. On the other hand, flow cytometry is present in most haematology laboratories and should the author’s findings be confirmed also by other centres, it could result in a very useful prognostic indicator, easily employed for most laboratories.

Overall I found the manuscript a little lengthy perhaps, in particular the discussion. However I would not at this stage suggest changes to shorten it. English editing however would be desirable.

The only additional point I would like for the authors to include is the following:

I think it would be relevant to place in the discussion that at least one previous large study reported CD2 as expressed on majority of T-LBL (and T-ALL) paediatric cases (Patel, J. L., Smith, L. M., Anderson, J., Abromowitch, M., Campana, D., Jacobsen, J., ... & Perkins, S. L. (2012). The immunophenotype of T‐lymphoblastic lymphoma in children and adolescents: a C hildren's O ncology G roup report. British journal of haematology159(4), 454-461). Other repot also CD2 negative cases, albeit in smaller cohorts (for example Kato, H., Yamamoto, K., Kodaira, T., Higuchi, Y., Yamamoto, H., Saito, T., ... & Kinoshita, T. (2018). Immunophenotypic analysis of adult patients with T-cell lymphoblastic lymphoma treated with hyper-CVAD. Hematology23(2), 83-88.). So, place other reports on frequency of CD2 expression in T-LBL in context with the author’s findings. Additionally, suggest consider to put a flow plot of a CD2 negative case, if not in main manuscript, perhaps as supplemental.

All the best,

Reviewer 4 Report

The authors show that the lack of CD2 expression prevailing in early /proT T-LBL subtype, and age over 35 years are powerful independent prognostic factor influencing OS and the risk of relapse. This is an interesting, but not a very original finding and all the paper, particularly in the Discussion section is too long and should be shortened in order to acquire more synthesis and clarity.

Author Response

There were 3 reviewers

Round 2

Reviewer 4 Report

No further criticisms.